# Dietary calcium intake among Iranian adults: Iranian Multicenter Osteoporosis Study (IMOS-2021)

**Arash Ghazbani**[1,2], **Nazli Namazi**[3], **Mohammad Javad Mansourzadeh**[2], **Kazem Khalagi**[2,4], **Navid Ostovar**[5], **Mahnaz Sanjari**[2], **Nekoo Panahi**[6,7], **Farideh Razi**[8], **Fatemeh Hajivalizadeh**[9], **Sepideh Hajivalizadeh**[2], **Elahe Hesari**[2], **Amirhossein Aghakhani**[1,2], **Farshad Farzadfar**[10], **Alireza Raiesi**[11], **Noushin Fahimfar**[1,2]*, **Afshin Ostovar**[2,7]*

1 Department of Epidemiology and Biostatistics, School of Public Health, Tehran University of Medical Sciences, Tehran, Iran, 2 Osteoporosis Research Center, Endocrinology and Metabolism Clinical Sciences Institute, Tehran University of Medical Sciences, Tehran, Iran, 3 Diabetes Research Center, Endocrinology and Metabolism Clinical Sciences Institute, Tehran University of Medical Sciences, Tehran, Iran, 4 Obesity and Eating Habits Research Center, Endocrinology and Metabolism Clinical Sciences Institute, Tehran University of Medical Sciences, Tehran, Iran, 5 Food and Beverage Safety Research Center, Urmia University of Medical Sciences, Urmia, Iran, 6 Metabolic Disorders Research Center, Endocrinology and Metabolism Cellular and Molecular Institute, Tehran University of Medical Sciences, Tehran, Iran, 7 Endocrinology and Metabolism Research Center, Endocrinology and Metabolism Clinical Sciences Institute, Tehran University of Medical Sciences, Tehran, Iran, 8 Metabolomics and Genomics Research Center, Endocrinology and Metabolism Molecular-Cellular Sciences Institute, Tehran University of Medical Sciences, Tehran, Iran, 9 Center for Non-Communicable Disease Control & Prevention, Deputy of Public Health, Ministry of Health and Medical Education, Tehran, Iran, 10 Non-Communicable Diseases Research Center, Endocrinology and Metabolism Population Sciences Institute, Tehran University of Medical Sciences, Tehran, Iran, 11 Department of Internal Medicine, School of Medicine, Tehran University of Medical Sciences, Tehran, Iran

* aostovar@sina.tums.ac.ir (AO); nfahimfar@gmail.com (NF)

**Data Availability Statement:** The data underlying this study are not publicly available due to sensitive patient information and ethical restrictions imposed

## Abstract

### Background

Adequate dietary consumption of calcium is crucial in the preservation of bone health and the prevention of osteoporosis. This study investigated the prevalence of insufficient dietary calcium intake among individuals aged $\geq$50 years in Iran.

### Methods

We analyzed data from the Iranian Multicenter Osteoporosis Study (IMOS-2021). Participants aged 50 years and older completed a 168-item food frequency questionnaire. Insufficient dietary calcium intake was characterized as a daily calcium intake of <1000 mg for men aged 50–70 years, and <1200 mg for men over 70 years and women over 50 years and older. *Stata v17* statistical software facilitated a survey set analysis to estimate the population's mean and median dietary calcium intake and the prevalence of insufficient dietary calcium intake.

by Endocrinology and Metabolism Research Institute (EMRI), but they can be accessed upon reasonable request. Researchers who wish to access the data for replication or other academic purposes may contact the corresponding authors or the Osteoporosis Research Center at EMRI via email at emri-osteoporosis@tums.ac.ir. Requests should include a brief description of the intended use of the data. Access to the data will be granted in accordance with institutional guidelines and ethical approvals, and may require the completion of a data sharing agreement to ensure compliance with confidentiality and data protection standards.

**Funding:** The Iranian Multicenter Osteoporosis Study (IMOS) received joint funding from the Iranian National Institute for Medical Research Development (NIMAD) and the Endocrinology and Metabolism Research Institute (EMRI) at Tehran University of Medical Sciences. Some of the researchers from the present study are affiliated to the EMRI; However, the grant body is independent to the researchers and funders had no role in study design, data collection and analysis, decision to publish, or preparation of the manuscript.

**Competing interests:** The authors have declared that no competing interests exist.

## Results

The study included 1450 participants with a mean age of 60.7±7.9 years. The estimated mean dietary calcium intake in Iran was 1062.7 mg/day (95% CI: 1029.6–1095.8), with a median intake of 943.5 mg/d (95% CI: 910.5–976.4). The prevalence of insufficient dietary calcium intake in Iran was estimated to be 62.9% (95% CI: 60.0%-65.7%). Notably, the prevalence was higher among women at 75.5% (95% CI: 71.9%-78.8%), compared to men at 47.8% (95% CI: 43.4%-52.3%) with a significant difference (P<0.001). In age-related findings, individuals aged 65 years and older had a higher prevalence of insufficient intake, at 69.0% (95% CI: 63.9%-74.0%), versus those under 65 years, at 60.3% (95% CI: 56.9%-63.8%), with this difference being statistically significant (P = 0.007). Furthermore, a significant inverse relationship was identified between both educational years and socioeconomic status and the prevalence of insufficient dietary calcium intake (Ps for trends<0.001).

## Conclusion

Our findings revealed a significant prevalence of insufficient dietary calcium intake in women and those aged 65 and older. We advocate for targeted public health strategies to ensure sufficient dietary calcium intake across these populations.

## Background

Calcium is a fundamental mineral that plays crucial roles in the skeletal, cardiovascular, endocrine, and neurological systems, thereby demonstrating its indispensability [1]. The majority, around 99%, of the overall calcium content in the human body is primarily localized within the skeletal system, playing the role of imparting rigidity and structural integrity to the bones. The remaining portion of calcium, constituting a fraction of the total calcium content, actively engages in various metabolic processes. These processes encompass vascular and muscle contraction, transmission within the nervous system, transmembrane transport, activation of enzymes, and facilitation of hormonal functions [2–4].

The lack of a reliable biomarker for evaluating calcium intake in population surveys necessitates the utilization of calcium intake data as a means of assessing the sufficiency of calcium levels at the population level. Dietary reference guidelines set by several agencies are used to measure the prevalence of insufficient calcium intake in a population by comparing the calcium intake with the age-specific estimated average requirement [2]. A significant proportion of research studies examining the long-term consequences of insufficient dietary calcium intake primarily focus on its impact on bone health. Calcium intake represents one among several contributing factors that influence the development of peak bone mass during growth and the maintenance of bone mass in adulthood [5]. Consequently, obtaining adequate amounts of bioavailable calcium throughout life is of growing importance as a major component in osteoporosis prevention.

Dairy products, such as milk, cheese, sour milk, and yogurt represent some of the most abundant natural reservoirs of calcium, as well as being significant sources of protein, potassium, and magnesium [6]. Compared with plant-based foods, bioavailability of calcium is also relatively high in dairy foods. However, the high costs and limited shelf-life of dairy products make them less accessible to many households living in lower- and middle-income countries [7]. Studies have demonstrated that out of the estimated 3.5 billion individuals who are at risk

of experiencing insufficient calcium intake, approximately 90% of them reside in Africa and Asia [8]. Besides, several countries in South and East Asia, have notably lower average dietary calcium intakes compared to Western nations, with some countries even falling below 500 mg/day. In contrast, Northern European countries stand out as the only nations where national calcium intake exceeds 1000 mg/day, highlighting a substantial disparity in calcium consumption between these regions [5].

In the context of Iran, there has been a relative scarcity of studies investigating dietary calcium intake. Findings from the Mashhad Stroke and Heart Atherosclerosis Disorder (MASHAD) study conducted in 2017 revealed that the average dietary calcium intake among the Iranian population was 862.4 mg/day for men, and 864.2 mg/day for women [9]. According to a report from the Bushehr Elderly Health (BEH) program published in 2021, approximately 92% of the participants had dietary calcium intake below 1000 mg/day [10]. Both investigations employed a 24-hour dietary recall method to evaluate dietary consumption with a focus on assessing calcium intake within an urban locale.

The present study aimed to address a critical gap in the existing literature by examining dietary calcium intake among Iranian adults aged 50 years and older. Our research breaks new ground by utilizing a nationally representative sample from Iran, enabling us to assess not only the prevalence of insufficient calcium intake but also to identify the primary sources of calcium in this specific population. By employing a comprehensive food frequency questionnaire (FFQ), we aimed to capture dietary patterns with greater accuracy compared to past studies. The findings of this study have significant implications for public health and nutrition, as they provide essential insights into dietary deficiencies and inform targeted interventions to improve calcium intake among older adults in Iran. This research has the potential to shape national dietary guidelines and public health policies, ultimately contributing to the prevention of osteoporosis and other calcium-related health issues in the aging population.

## Materials and methods

### Study population and design

Iranian Multicenter Osteoporosis Study (IMOS-2021) is a population-based cross-sectional survey that has been conducted in Iran to estimate the prevalence of osteoporosis, sarcopenia, and their possible related risk factors in the country. The full protocol of the IMOS-2021 has been previously documented elsewhere [11]. The target population of the IMOS comprises all Iranian men and women aged 50 years and over residing in the country. Since the main objective of the IMOS was measuring the prevalence of osteoporosis and sarcopenia, subjects with a weight more than 150 kg, cystic fibrosis, pregnant women, subjects with prohibition to measure Bone Mineral Density (BMD) (such as bilateral hip replacement), and those with history of recent contrast for imaging were excluded. The written informed consent obtained from all participants.

### Sampling method

IMOS stratified 31 provinces of Iran into 5 strata based on the distribution of their potential risk factors for osteoporosis and randomly selected one or two provinces from each stratum. In random selection, the most populous provinces in each stratum were given a higher chance of recruitment. Accordingly, the provinces of Tehran, Isfahan, Mazandaran, West Azerbaijan, North Khorasan, Fars, Kermanshah, and Khuzestan were selected for the study. Then, people aged 50 years and older that were recruited in the 8th National Survey of Non-Communicable Diseases (NCDs) Risk Factors (STEPs-2021) [12] were invited to participate in IMOS. The recruitment phase commenced on October 2nd, 2021, and ended on January 4th, 2023.

## Measurements

To understand the characteristics of our sample, demographic data (age, gender, areas of residence, education, and socioeconomic status) collected through a self-reported questionnaire. Socioeconomic status was defined based on individuals' asset data and Principal Component Analysis (PCA) was conducted to generate a wealth index [13]. Scores derived from PCA were divided into quintiles. A higher quintile indicates better socioeconomic status.

In order to evaluate the dietary minerals of the participants, a 168-item semi-quantitative FFQ was used. The questionnaire had already been validated in Iran [14]. It contains a list of foods (with standard serving sizes) commonly consumed by Iranians. IMOS participants were requested to report their frequency of consumption of a given serving of each food item during the past year, on a daily, weekly, monthly, or yearly basis. To ensure the validity and reliability of the data collection, we employed trained interviewers who administered the FFQ, guiding participants through the process and clarifying any questions they had about the survey. The frequency reported for each food item was then converted to a daily intake. The weight of seasonal items, such as some vegetables, was calculated based on the number of seasons that each food was available. The amount and frequency of dietary intake documented within the FFQ were transcribed into the *Nutritionist IV* software subsequent to the conversion of individual food items to daily consumption. Subsequently, an analysis of the dietary components, including calcium, was conducted.

The prevalence of insufficient dietary calcium intake was determined based on the Dietary Reference Intakes (DRIs) developed by the Food and Nutrition Board (FNB) at the National Academies of Sciences, Engineering, and Medicine. DRI is the general term for a set of reference values used for planning and assessing nutrient intakes of healthy individuals. The insufficient dietary intake of calcium was defined as daily calcium intake of <1000 mg/day for men aged 50–70 years, <1200 mg/day for men aged 70 years and older, and <1200 mg/day for women aged 50 years and older [15]. The share of the food basket was conceptualized as the proportion of specific food items within the diet or consumption patterns of the study population.

## Statistical analysis

We examined the deviation of distribution of data on calcium intake from the normal distribution using histogram and Shapiro-Wilk test (P-value more than 0.05 was determined as normal distribution). Moreover, we estimated the prevalence of insufficient dietary calcium intake with 95% Confidence Interval (CI) at the national level and by gender, age groups (below and above 65 years), the area of residence (urban and rural), education (no education, diploma or less, and college and more), and socioeconomic status (poorest, second, middle, fourth, and richest).

We applied a survey set analysis to obtain the prevalence estimates. The following weights were used for this purpose:

i.  The weight of the correcting differences in the age, gender, and area of residence distribution of the study sample with their distribution in the population of the province (post-stratification weight). This weight is the inverse of the ratio of the number of samples determined for each age, gender, and area of residence category by the population of that category in each selected province [11];

ii. the weight of the responding which was used to correct the effect of non-responses on the prevalence estimates [16]. Since the variables age, gender, area of residence, and stratum of residence are associated with participation in the study, this weight was obtained by

dividing the number of determined samples by the number of participants (inverse of the probability of responses) in each of the age, gender, and area of residence categories for each province;

iii. the sampling weight which was calculated by dividing the number of the population of each stratum by the sample size of that stratum [17].

Statistical analysis and mapping were performed using *Stata v17* (Stata Corp, College Station, TX, USA). *Stata*'s facilities for survey data analysis are centered around the *svy* prefix command. After identification of the survey design characteristics with the *svyset* command, we prefixed the estimation commands in our data analysis with *svy*. In order to visualize the prevalence of insufficient dietary calcium intake within each stratum, *spmap* command was utilized. Additionally, the primary sources of calcium in the participants' food basket (common food items) were identified through the findings obtained from *Nutritionist IV*.

### Ethical considerations

The ethics committee of the Iranian National Institute for Medical Research Development (NIMAD) authorized the Iranian Multicenter Osteoporosis Study (IMOS) on February 2, 2019, with the Approval ID: IR.NIMAD.REC.1398.056. All participants provided written informed consent before joining the study.

## Results

In this research, a comprehensive sample of 1,450 individuals was recruited. The mean participants' age was 60.7±7.9 years, with a range from 50 to 94 years old; female identification comprised 54.6% of the participants. Notably, 70.1% of the participants fell below the age of 65. Additionally, 74.6% of the subjects resided in urban settings. Table 1 provided a detailed summary of the baseline characteristics of the participants.

As per findings from the present research, the distribution of calcium intake among Iranian adults exhibited a right-skewed pattern (Fig 1), which its deviation from normal distribution was supported by the results of the Shapiro-Wilk test (P<0.001). The central tendency and variability measurements of dietary calcium intake presented in S1 Table.

Based on our findings, the weighted median ± interquartile range and mean ± standard deviation calcium intake among Iranian adults estimated to be 943.5 ± 620.2 mg/day (95% CI: 910.5–976.4) and 1062.7 ± 542.2 mg/day (95% CI: 1029.6–1095.8), respectively. The weighted median dietary calcium intake was 1067.2 mg/day (95% CI: 1007.6–1126.8) and 855.2 mg/day (95% CI: 820.5–890.0) among men and women, and it was 965.4 mg/day (95% CI: 929.1–1001.7) and 869.8 mg/day (95% CI: 806.0–933.6) among the participants aged <65 and ≥65 years old, respectively. Moreover, the weighted median dietary calcium intake was 957.9 mg/day (95% CI: 915.1–1000.8) and 891.4 mg/day (95% CI: 820.6–962.2) among urban and rural populations, respectively. The weighted mean dietary calcium intake was 1180.6 mg/day (95% CI: 1126.0–1235.2) and 964.4 mg/day (95% CI: 925.5–1003.3) among men and women, and it was 1088.2 mg/day (95% CI: 1046.2–1130.1) and 1002.3 mg/day (95% CI: 951.5–1053.1) among the participants aged <65 and ≥65 years, respectively. Moreover, the weighted mean dietary calcium intake was 1072.8 mg/day (95% CI: 1034.9–1110.7) and 1033.5 mg/day (95% CI: 965.1–1101.8) among urban and rural populations, respectively (S1 Table and Fig 2A).

The total weighted prevalence of insufficient dietary calcium intake estimated to be 62.9% (95% CI: 60.0%-65.7%). The prevalence of insufficient dietary calcium intake within each stratum illustrated in Fig 3. The prevalence of insufficient dietary calcium intake was 47.8% (95% CI: 43.4%-52.3%) and 75.5% (95% CI: 71.9%-78.8%) among men and women, respectively. It

**Table 1. Sociodemographic characteristics of the participants contributed to the Iranian Multicenter Osteoporosis Study (IMOS-2021).**

| Variable | | Sufficient (n = 515) | Insufficient (n = 935) | P* |
|---|---|---|---|---|
| **Gender** | **Male** | 330 (50.1) | 328 (49.9) | **<0.001** |
| | **Female** | 185 (23.3) | 607 (76.7) | |
| **Age groups** | **<65** | 383 (37.8) | 631 (62.2) | **0.006** |
| | **≥65** | 131 (30.3) | 301 (69.7) | |
| **Area of residence** | **Rural** | 127 (34.6) | 240 (65.4) | 0.662 |
| | **Urban** | 387 (35.9) | 692 (64.1) | |
| **Education** | **No education** | 97 (27.9) | 251 (72.1) | **<0.001[†]** |
| | **Diploma or less** | 255 (35.4) | 466 (64.6) | |
| | **College and more** | 161 (42.8) | 215 (57.2) | |
| **SES quintiles** | **Poorest** | 66 (23.4) | 216 (76.6) | **<0.001[†]** |
| | **Second** | 99 (35.6) | 179 (64.4) | |
| | **Middle** | 106 (37.5) | 177 (62.5) | |
| | **Fourth** | 115 (40.6) | 168 (59.4) | |
| | **Richest** | 113 (41.7) | 158 (58.3) | |

Values are number (%), SES quintiles: Socioeconomic status quintiles

*P-value using proportion test

[†] P for trend using chi-square test

was 60.3% (95% CI: 56.9%-63.8%) and 69.0% (95% CI: 63.9%-74.0%) among the participants younger and older than 65, respectively. The total weighted prevalence of insufficient dietary calcium intake in urban and rural populations estimated to be 62.1% (95% CI: 58.7%-65.5%) and 65.3% (95% CI: 59.9%-70.8%), respectively. A significant inverse correlation observed between the prevalence of insufficient dietary calcium intake and years of education (p for trend<0.001) (S2 Table and Fig 2B). This inverse correlation also observed between the prevalence of insufficient dietary calcium intake and socioeconomic status (p for trend<0.001).

Based on our findings, dairy products constituted the largest proportion of calcium sources within the dietary composition of Iranian adults, accounting for 53.2%, followed by grains (19.3%). Additionally, "bread" emerged as the primary source of calcium intake, representing 16.0% of the Iranian food basket, followed by "normal yogurt" (13.0%) and "yogurt drink" (12.0%) (Table 2).

## Discussion

In the present study, we conducted a comprehensive analysis of dietary calcium intake among Iranian adults aged 50 years and older. The weighted median and mean dietary calcium intakes were ascertained to be 943.5 mg/day and 1062.7 mg/day, respectively. Furthermore, our research determined that the overall weighted prevalence of insufficient dietary calcium intake in this demographic is 62.9%. Additionally, it was observed that dairy products represent the most significant source, contributing to 53.2% of the total calcium intake within the dietary patterns of the studied population.

The data from this investigation revealed a mean dietary calcium intake that aligns with figures reported in Northern European nations [5]. Nevertheless, the high prevalence of insufficient dietary calcium intake and the comparatively lower median daily intake observed in our study suggest that the calculated mean could be skewed by extreme values and should be taken into consideration when interpreting the findings [18]. Comparative analysis with prior research conducted in 2001 and 2017 in Iran indicates an elevation in mean dietary calcium

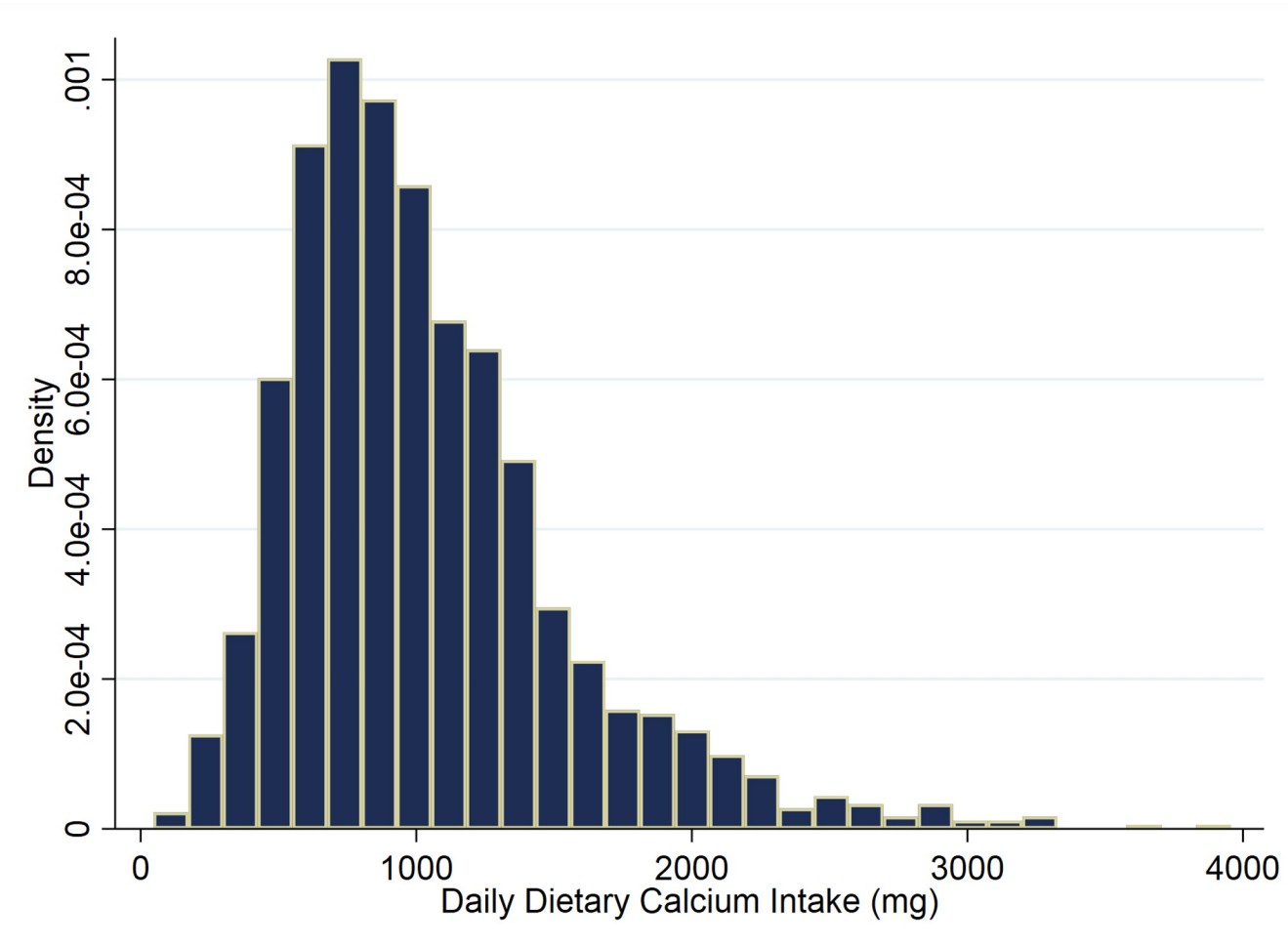

**Fig 1. Distribution of dietary calcium intake among Iranian older adults: Iranian Multicenter Osteoporosis Study (IMOS-2021).**

intake in the current study [9, 19]. Such discrepancies may stem from methodological differences in dietary data collection and temporal gaps between the studies [20, 21]. It is pertinent to note that the IMOS represents a more contemporary approach, encompassing a nationally representative sample of the Iranian population aged 50 years and older and employing a standardized FFQ. Consequently, the estimates derived from our study potentially offer a more reliable picture of dietary calcium intake, in contrast to regional studies that utilized varied dietary recalls or FFQs of differing lengths.

The current study elucidates a substantial prevalence of insufficient dietary calcium intake among the studied population. This prevalence markedly exceeds the estimates reported by Beal et al., and the Food Balance Sheets (FBSs-2011) provided by the United Nations Food and Agriculture Organization (FAO), which posited a prevalence ranging from 40% to 50% [22]. Such a discernible escalation in the prevalence of insufficient calcium intake since 2011 in Iran is a cause for considerable concern. Nonetheless, this phenomenon is not isolated to Iran; it reflects a global health challenge. For instance, Sales et al. have documented prevalence of insufficient calcium intake in Brazil falls between 77.9% and 99.8% [23]. Similarly, Kumssa et al. have identified that approximately 3.5 billion individuals globally are at risk for insufficient calcium intake, with nearly 90% residing in Africa and Asia [8]. In this regard, the

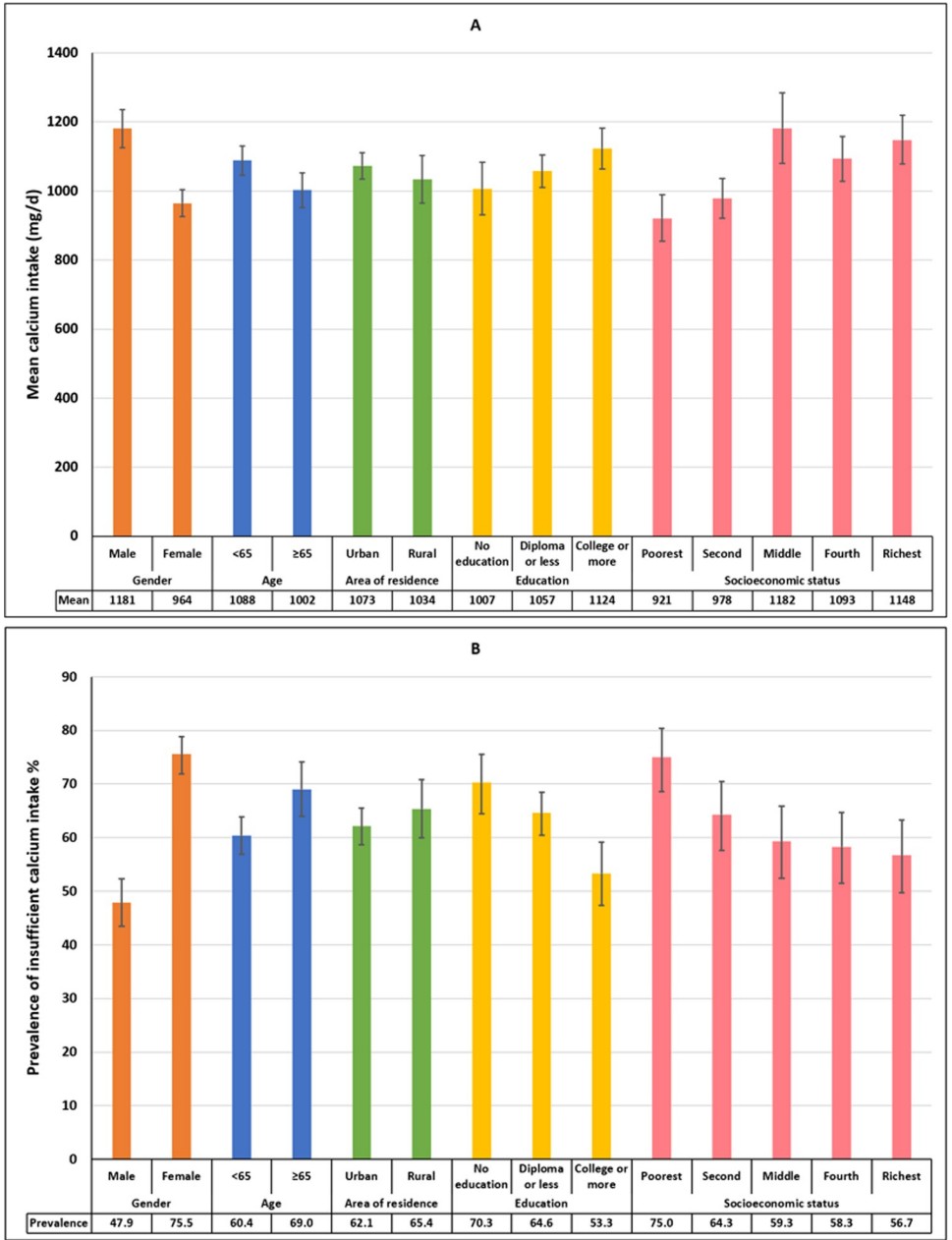

**Fig 2. A**: Mean dietary calcium intake and **B**: the prevalence of insufficient dietary calcium intake among Iranian older adults: Iranian Multicenter Osteoporosis Study (IMOS-2021).

primary factors contributing to low calcium intake have been identified as limited availability of calcium-rich foods, traditional dietary practices, food insecurity, and gender discrimination. Food insecurity and poor purchasing power due to economic challenges also limit dairy products consumption by most of the population in lower- and middle-income countries [2]. Based on the findings of the present study, these factors may also contribute to the high prevalence of insufficient calcium intake in Iran. Given the critical role of calcium in maintaining bone health and the potential for prolonged calcium deficiency to precipitate reduced bone mineral density, heightened susceptibility to fractures, and other musculoskeletal disorders

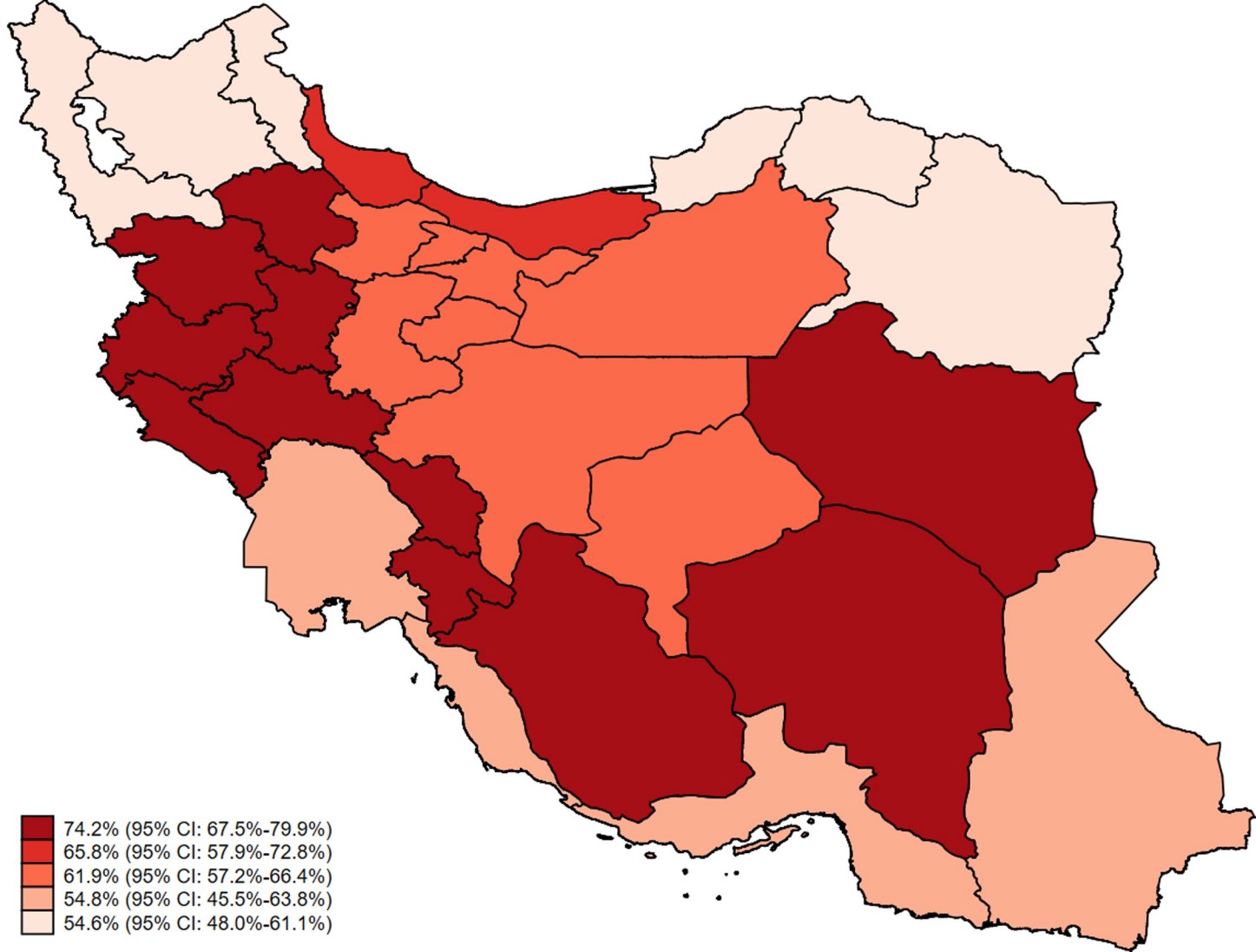

74.2% (95% CI: 67.5%-79.9%)
65.8% (95% CI: 57.9%-72.8%)
61.9% (95% CI: 57.2%-66.4%)
54.8% (95% CI: 45.5%-63.8%)
54.6% (95% CI: 48.0%-61.1%)

**Fig 3. The prevalence of insufficient dietary calcium intake among Iranian older adults by each stratum of the Iranian Multicenter Osteoporosis Study (IMOS-2021) (the map depicted originally by the authors using *Stata v17*).**

[24–26], it is imperative to implement strategic interventions and continuous monitoring of calcium intake.

Our study identified a global trend of lower dietary calcium intake among women and older individuals, not confined to the Iranian population. Balk et al.'s systematic review supports this, demonstrating a median intake ratio of 0.9 between genders across 42 countries. This is particularly concerning for women, as sufficient calcium intake is crucial for preventing osteoporosis, a condition that disproportionately affects women more than men [27]. Consistent with our findings, the mentioned research indicates that older individuals tend to have lower dietary calcium intake. This suggests that the pattern of lower calcium intake among older individuals is not specific to the Iranian population but is observed in various countries worldwide. This trend may be partly attributed to lactose intolerance in the elderly making it difficult for them to digest lactose found in milk and other dairy products [28]. Contrarily, our results revealed no significant difference in calcium intake between urban and rural residents,

**Table 2. Main sources of dietary calcium intake in the food basket of Iranian adults: Iranian Multicenter Osteoporosis Study (IMOS-2021).**

| Rank | Calcium source (food items) | Calcium (mg) per 1 mg of food item | Share of the food basket % | Cumulative percent % |
|------|------------------------------|-------------------------------------|-----------------------------|-----------------------|
| 1 | Bread (in different forms) | 0.7 to 1.5 | 16.0 | 16.0 |
| 2 | Normal yogurt | 1.6 | 13.0 | 29.0 |
| 3 | Yogurt drink (Doogh) | 1.6 | 12.0 | 41.0 |
| 4 | Cheese | 4.8 | 10.7 | 51.7 |
| 5 | High-fat yogurt | 1.5 | 7.1 | 58.8 |
| 6 | High-fat milk | 1.1 | 5.1 | 63.9 |
| 7 | Low-fat milk | 1.1 | 3.2 | 67.1 |
| 8 | Orange | 0.4 | 3.0 | 70.1 |
| 9 | Rice | 0.1 | 2.3 | 72.4 |
| 10 | Fried onion | 0.8 | 1.6 | 74.0 |
| **Rank** | **Calcium source (food groups)** | **Share of the food basket %** | | **Cumulative percent %** |
| 1 | Dairy products | 53.2 | | 53.2 |
| 2 | Grains | 19.3 | | 72.5 |
| 3 | Vegetables | 8.4 | | 80.9 |
| 4 | Fresh fruits | 7.9 | | 88.8 |
| 5 | Sweet products and sugars | 2.8 | | 91.6 |

challenging findings from low-resource countries where rural intake was significantly lower. This may reflect a lifestyle convergence between rural and urban areas, as indicated by similar health indicators like obesity and diabetes rates [29–31].

Furthermore, our analysis revealed a robust association between socioeconomic status and educational attainment with calcium consumption, corroborating the findings of prior studies [32, 33]. Considering the prevalent under recognition of calcium's vital role in health, targeted educational programs have the potential to substantially enhance calcium intake [34]. Additionally, social initiatives aimed at poverty alleviation may contribute to diminishing disparities in access to calcium-rich foods, thereby addressing nutritional inequities. These initiatives, which may include food assistance programs and subsidies for healthier food options, help ensure that economically disadvantaged populations have better access to essential nutrients [35]. As a result, these initiatives can significantly contribute to reducing nutritional inequities, leading to better overall health outcomes.

Our study revealed that dairy products significantly contribute to the calcium intake of Iranian adults, making up about half of their total calcium consumption. However, economic sanctions have affected the affordability and accessibility of these essential food groups [36]. Rising prices for milk, yogurt, and cheese, coupled with reduced production and distribution, pose challenges for many Iranian families, particularly those in vulnerable socioeconomic situations, in obtaining adequate dietary calcium [37]. This might also contribute to the high prevalence of insufficient dietary calcium intake in the studied population.

The IMOS provides recent population-based evidence by utilizing a nationally representative sample of adults aged 50 years and older, encompassing both genders and including urban and rural populations in Iran. The study employed the proper FFQ as a standard tool for measuring dietary calcium intake. Nevertheless, there are some limitations to our research that should be considered. While IMOS focused on adults aged 50 years and older, it is important to recognize that dietary patterns and calcium intake levels can vary significantly across different age groups and demographic characteristics. Therefore, caution is advised when applying our findings to other subsets of the population. It is worth noting that the FFQ largely depends on participants' ability to recall their dietary intake, which can be particularly challenging

among older adults. This reliance on memory could introduce measurement errors that may either overestimate or underestimate actual dietary calcium intake.

## Conclusion

The Iranian population exhibits a markedly high prevalence of insufficient dietary calcium intake, notably among women (who are already at higher risk of osteoporosis) and the elderly ($\geq$65 years). The potential severe health implications of such deficiencies, particularly for older adults, necessitate urgent educational initiatives to underscore the critical role of calcium. There is an imperative to disseminate knowledge regarding established nutritional guidelines and preventative strategies to mitigate this public health concern.

Healthcare policymakers should address calcium insufficiencies by promoting better nutritional intake. This includes educating the public on balanced diets, advocating for more consumption of calcium-rich foods (milk, cheese, yogurt, dairy), and introducing public health campaigns and primary prevention programs. Offering subsidies for calcium-fortified foods, improving food labeling, and increasing access to affordable, nutrient-dense foods in all areas should also considered. These actions would help ensure sufficient calcium intake, particularly for women, reducing the risk of deficiencies such as osteoporosis.

## Supporting information

**S1 Table. Central tendency measures of dietary calcium intake: Iranian Multicenter Osteoporosis Study (IMOS-2021).**
(DOCX)

**S2 Table. Weighted prevalence of insufficient dietary calcium intake: Iranian Multicenter Osteoporosis Study (IMOS-2021).**
(DOCX)

## Acknowledgments

We are grateful for the cooperation of all those involved.

## Author Contributions

**Conceptualization:** Arash Ghazbani, Noushin Fahimfar, Afshin Ostovar.

**Data curation:** Arash Ghazbani, Mohammad Javad Mansourzadeh, Navid Ostovar.

**Formal analysis:** Arash Ghazbani, Nazli Namazi, Mohammad Javad Mansourzadeh, Kazem Khalagi, Navid Ostovar, Amirhossein Aghakhani.

**Investigation:** Farideh Razi.

**Methodology:** Arash Ghazbani, Nazli Namazi, Kazem Khalagi, Noushin Fahimfar, Afshin Ostovar.

**Software:** Arash Ghazbani, Mohammad Javad Mansourzadeh, Navid Ostovar, Amirhossein Aghakhani.

**Supervision:** Noushin Fahimfar, Afshin Ostovar.

**Visualization:** Arash Ghazbani.

**Writing – original draft:** Arash Ghazbani, Nazli Namazi, Kazem Khalagi, Nekoo Panahi, Noushin Fahimfar, Afshin Ostovar.

**Writing – review & editing:** Arash Ghazbani, Nazli Namazi, Mohammad
Javad Mansourzadeh, Kazem Khalagi, Navid Ostovar, Mahnaz Sanjari, Nekoo Panahi,
Farideh Razi, Fatemeh Hajivalizadeh, Sepideh Hajivalizadeh, Elahe Hesari,
Amirhossein Aghakhani, Farshad Farzadfar, Alireza Raiesi, Noushin Fahimfar,
Afshin Ostovar.

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
