## [Decision Letter · Decision Letter 0]

22 Jul 2024

PONE-D-24-25483Dietary calcium intake among Iranian adults: Iranian Multicenter Osteoporosis Study (IMOS-2021)PLOS ONE

Dear Dr. Ostovar,

Thank you for submitting your manuscript to PLOS ONE. After careful consideration, we feel that it has merit but does not fully meet PLOS ONE’s publication criteria as it currently stands. Therefore, we invite you to submit a revised version of the manuscript that addresses the points raised during the review process.

We look forward to receiving your revised manuscript.

Kind regards,

Melissa Orlandin Premaor, M.D., Ph.D

Academic Editor

PLOS ONE

Journal Requirements:

   "The Iranian Multicenter Osteoporosis Study (IMOS) received joint funding from the Iranian National Institute for Medical Research Development (NIMAD) and the Endocrinology and Metabolism Research Institute (EMRI) at Tehran University of Medical Sciences. "

5. In this instance it seems there may be acceptable restrictions in place that prevent the public sharing of your minimal data. However, in line with our goal of ensuring long-term data availability to all interested researchers, PLOS’ Data Policy states that authors cannot be the sole named individuals responsible for ensuring data access (http://journals.plos.org/plosone/s/data-availability#loc-acceptable-data-sharing-methods).

7. We note that Figure 3 in your submission contain map/satellite images which may be copyrighted. All PLOS content is published under the Creative Commons Attribution License (CC BY 4.0), which means that the manuscript, images, and Supporting Information files will be freely available online, and any third party is permitted to access, download, copy, distribute, and use these materials in any way, even commercially, with proper attribution. For these reasons, we cannot publish previously copyrighted maps or satellite images created using proprietary data, such as Google software (Google Maps, Street View, and Earth). For more information, see our copyright guidelines: http://journals.plos.org/plosone/s/licenses-and-copyright.

a. You may seek permission from the original copyright holder of Figure 3 to publish the content specifically under the CC BY 4.0 license.  

Reviewers' comments:

Reviewer's Responses to Questions

**Comments to the Author**

1. Is the manuscript technically sound, and do the data support the conclusions?

Reviewer #1: Yes

Reviewer #2: Yes

2. Has the statistical analysis been performed appropriately and rigorously? 

Reviewer #1: Yes

Reviewer #2: Yes

3. Have the authors made all data underlying the findings in their manuscript fully available?

Reviewer #1: Yes

Reviewer #2: Yes

4. Is the manuscript presented in an intelligible fashion and written in standard English?

Reviewer #1: Yes

Reviewer #2: Yes

5. Review Comments to the Author

Reviewer #1: The national level food intake data by authors is commendable as currently actual food intake data is lacking in most countries.

IMOS being a study with >50 yrs population and mean age of 60.7 yrs the Calcium and FFQ based diet intake does not meet national populational representation, however it would add new information for this group only. The FFQ based diet analysis that too in old age is challenging interms of correct information.

However there are few suggestions for authors:

a) If possible the authors may work upon how we can measure the diet quality at national level.

b) Metrics that measures the role of various factors which define dietary choices.

c) Metrices which measure the food environment of food choices.

d) Measuring the health of food system (Overall) at country level.

Reviewer #2: The topic of dietary calcium intake is of significant importance for public health, and your study provides valuable insights into the nutritional challenges faced by older adults in Iran. Below are some detailed points for your consideration to enhance the clarity and impact of your paper.

0. The Grammar and flow of the manuscript needs special attention. (ex. incomplete sentences, capitalization, flow and coherence of paragraphs)

1. End the introduction with a stronger statement on the anticipated impact of your findings, emphasizing their significance in the broader context of public health and nutrition.

2. The last sentence of statistical analysis is incomplete: "The primary sources of calcium in the participants' food basket (common food items were identified through the findings obtained from Nutritionist IV."

3. Correct the capitalization of words (ex. "confidence Interval" to "Confidence Interval" and "STATA V17" to "Stata v17".)

4. I noticed that the study period spans from October 2nd, 2021, to January 4th, 2023, which is approximately 459 days. Could you please clarify whether this extended duration aligns with the cross-sectional study design? Typically, cross-sectional studies are conducted over shorter periods to capture a specific snapshot in time. It would be helpful to understand how the study design accommodates this extended timeline and ensures the integrity of the cross-sectional approach.

5. to be reproducible, the methodology should be stated more clearly:

a. Specify the threshold used for the Shapiro-Wilk test to determine normal distribution.

b. Provide a clearer explanation of how each type of weight (ex. post-stratification, responding, and sampling weight) was calculated and applied.

c. Include more details on the survey set analysis methodology to clarify how it was conducted.

In the discussion:

a. Provide more context when comparing global trends, specifying why the Iranian population might follow these trends or deviate from them.

b. Make clear, actionable policy recommendations based on your findings, such as interventions to improve calcium intake in at-risk populations. because it is still unclear what the main achievement of this study was.

6. PLOS authors have the option to publish the peer review history of their article (what does this mean?). If published, this will include your full peer review and any attached files.

Reviewer #1: **Yes: **Suresh Yadav

Reviewer #2: No

---

## [Author Response · Author response to Decision Letter 0]

17 Aug 2024

Dear editor and reviewers,

Thanks for your valuable comments. We believe that by editing the manuscript, our research is now suitable for publishing in PLOS ONE journal.

Our answers to your comments are highlighted in blue.

Journal Requirements:

Answer: Thank you for your attention. We have addressed your comments below.

Answer: We have double checked the style requirements and file naming. Please let us know if further change needed.

Answer: The funding information was removed from the main files of the manuscript. 

Answer: We have double checked the funding information which was provided in the editorial manager. Please let us know if further change needed.

4. Please state what role the funders took in the study. 

Answer: We added further explanations in this regard. The founding information is presented below:

The Iranian Multicenter Osteoporosis Study (IMOS) received joint funding from the Iranian National Institute for Medical Research Development (NIMAD) and the Endocrinology and Metabolism Research Institute (EMRI) at Tehran University of Medical Sciences. Some of the researchers from the present study are affiliated to the EMRI; However, the grant body is independent to the researchers and funders had no role in study design, data collection and analysis, decision to publish, or preparation of the manuscript. 

5. In this instance it seems there may be acceptable restrictions in place that prevent the public sharing of your minimal data. However, in line with our goal of ensuring long-term data availability to all interested researchers, PLOS’ Data Policy states that authors cannot be the sole named individuals responsible for ensuring data access. 

Answer: We edited the data availability statement according to your comment: The data underlying this study are not publicly available due to sensitive patient information and ethical restrictions imposed by Endocrinology and Metabolism Research Institute (EMRI), but they can be accessed upon reasonable request. Researchers who wish to access the data for replication or other academic purposes may contact the corresponding authors or the Osteoporosis Research Center at EMRI via email at emri-osteoporosis@tums.ac.ir. Requests should include a brief description of the intended use of the data. Access to the data will be granted in accordance with institutional guidelines and ethical approvals, and may require the completion of a data sharing agreement to ensure compliance with confidentiality and data protection standards.

Answer: The ethics statement has now transferred to the method section.

7. We note that Figure 3 in your submission contain map/satellite images which may be copyrighted. 

Answer: We changed the Figure 3 in order to comply journal’s copyright policy. The new map was manually created by the authors using Stata v17, a statistical software that allows users to generate custom visualizations. In order to visualize the prevalence of insufficient dietary calcium intake within each stratum, spmap command was utilized (we added this statement to the statistical analysis section). We confirm that the map is an original creation and does not reproduce or copy any existing copyrighted material. We obtained the base map and data from https://data.humdata.org/m/dataset/cod-ab-irn? 

License link: https://data.humdata.org/faqs/licenses

9. Please include captions for your Supporting Information files at the end of your manuscript, and update any in-text citations to match accordingly. 

Answer: We added the captions of our supplementary materials, figures, and tables to the end of the main file of the manuscript.

Reviewer #1:

The national level food intake data by authors is commendable as currently actual food intake data is lacking in most countries.

IMOS being a study with >50 yrs population and mean age of 60.7 yrs the Calcium and FFQ based diet intake does not meet national populational representation, however it would add new information for this group only. The FFQ based diet analysis that too in old age is challenging in terms of correct information.

However, there are few suggestions for authors:

a) If possible, the authors may work upon how we can measure the diet quality at national level.

b) Metrics that measures the role of various factors which define dietary choices.

c) Metrices which measure the food environment of food choices.

d) Measuring the health of food system (Overall) at country level.

Answer: 

Thanks for your valuable comments. we found your ideas quite exciting and aligned with our upcoming research endeavors. 

As we have clarified in the manuscript, this study primarily focused on dietary calcium intake among Iranian adults, and as you mentioned, we believe that our research is only representative for population aged 50 years and over. However, there is few nation-wide, population-based research reporting dietary calcium intake in Iran. Therefore, our study would provide useful information for this purpose. 

Regarding FFQ, we agree that it has serious limitations for diet analysis especially in elderly. However, it is still a common method for measuring dietary patterns in large epidemiological studies of diet and health especially in low resource settings and it was the most valid and reliable method that we had for our purpose. 

Reviewer #2:

The topic of dietary calcium intake is of significant importance for public health, and your study provides valuable insights into the nutritional challenges faced by older adults in Iran. Below are some detailed points for your consideration to enhance the clarity and impact of your paper.

 0. The Grammar and flow of the manuscript needs special attention. (ex. incomplete sentences, capitalization, flow and coherence of paragraphs). 

Answer: 

Thanks for your valuable comment. We have double checked and edited the main file of the manuscript in this regard. Please let us know if there are any specific issues regarding grammar or other problems.

1. End the introduction with a stronger statement on the anticipated impact of your findings, emphasizing their significance in the broader context of public health and nutrition.

Answer: 

Thank you for your comment. We edited the last paragraph of the introduction to be stronger and indicated the implication of the study findings considering their potential public health impact. 

2. The last sentence of statistical analysis is incomplete: "The primary sources of calcium in the participants' food basket (common food items were identified through the findings obtained from Nutritionist IV."

Answer: 

Thanks for your attention. The sentence was edited.

3. Correct the capitalization of words (ex. “confidence Interval” to “Confidence Interval” and “STATA V17” to “Stata v17”.) 

Answer: 

Thanks for your attention. The capitalizations were edited.

4. I noticed that the study period spans from October 2nd, 2021, to January 4th, 2023, which is approximately 459 days. Could you please clarify whether this extended duration aligns with the cross-sectional study design? Typically, cross-sectional studies are conducted over shorter periods to capture a specific snapshot in time. It would be helpful to understand how the study design accommodates this extended timeline and ensures the integrity of the cross-sectional approach. 

Answer: 

Thank you for your comment. This study is a nationwide research endeavor, necessitating the establishment of specific infrastructures to ensure the validity and reliability of our results. Since the primary objective of IMOS was to assess the prevalence of osteoporosis in Iran, this included the provision of valid and calibrated DXA devices, as well as the recruitment and training of personnel. Furthermore, the data collection process extended beyond our expectations due to logistical challenges. The inclusion of samples from the rural population, coupled with the significant distances between these areas and the data collection centers, which were primarily located in large cities, contributed to the prolonged data collection period. Additionally, the data collection phase was further extended due to disruptions caused by the COVID-19 pandemic. However, we believe that our study had still a cross sectional design because the study period was not too long to expect significant change in the variables under study. 

5. to be reproducible, the methodology should be stated more clearly:

Thanks for your advice. We addressed your comments accordingly. Please refer to the highlighted change in the manuscript. 

a. Specify the threshold used for the Shapiro-Wilk test to determine normal distribution. 

Answer: 

P-value more than 0.05 was determined as normal distribution. This statement was added to the first paragraph of the statistical analysis section.

b. Provide a clearer explanation of how each type of weight (ex. post-stratification, responding, and sampling weight) was calculated and applied. 

Answer: 

In this regard, more information was added to the statistical analysis section. Further information was explained in the protocol paper which was cited in the method section.

c. Include more details on the survey set analysis methodology to clarify how it was conducted. 

Answer: 

We added more details about the survey set analysis to the last paragraph of the statistical analysis section.

6. In the discussion:

a. Provide more context when comparing global trends, specifying why the Iranian population might follow these trends or deviate from them.

Answer: 

We provided more context in paragraph 3 of the discussion. We have also addressed the primary reasons for the increasing prevalence of insufficient calcium intake among the Iranian population, in line with global trends in subsequent paragraphs.

b. Make clear, actionable policy recommendations based on your findings, such as interventions to improve calcium intake in at-risk populations. because it is still unclear what the main achievement of this study was. 

Answer: 

Thanks for your attention. We provided more detail in the conclusion section.

---

## [Editor Report · Decision Letter 1]

23 Aug 2024

Dietary calcium intake among Iranian adults: Iranian Multicenter Osteoporosis Study (IMOS-2021)

PONE-D-24-25483R1

Dear Dr. Ostovar,

We’re pleased to inform you that your manuscript has been judged scientifically suitable for publication and will be formally accepted for publication once it meets all outstanding technical requirements.

Kind regards,

Melissa Orlandin Premaor, M.D., Ph.D

Academic Editor

PLOS ONE
---

## [Editor Report · Acceptance letter]

28 Aug 2024

PONE-D-24-25483R1 

PLOS ONE

Dear Dr. Ostovar, 

I'm pleased to inform you that your manuscript has been deemed suitable for publication in PLOS ONE. Congratulations! Your manuscript is now being handed over to our production team.

Kind regards, 

on behalf of

Dr. Melissa Orlandin Premaor 

Academic Editor

PLOS ONE